# Perspectives on the Health Effects of Hurricanes: A Review and Challenges

**DOI:** 10.3390/ijerph18052756

**Published:** 2021-03-09

**Authors:** Samantha L. Waddell, Dushyantha T. Jayaweera, Mehdi Mirsaeidi, John C. Beier, Naresh Kumar

**Affiliations:** 1Case Western University, Cleveland, OH 44236, USA; slw129@case.edu; 2Miller School of Medicine, University of Miami, Miami, FL 33136, USA; djayawee@med.miami.edu; 3Division of Pulmonary, Allergy, Critical Care, Miller School of Medicine, University of Miami, Miami, FL 33136, USA; msm249@med.miami.edu; 4Division of Environmental Health Sciences, Department of Public Health Sciences, Miller School of Medicine, University of Miami, Miami, FL 33136, USA; jbeier@miami.edu; 5Division of Environmental Health, Department of Public Health Sciences, Miller School of Medicine, University of Miami, Miami, FL 33136, USA

**Keywords:** health impacts, hurricanes, environmental stressors, health impact persistence, time-lagged health effects of hurricanes

## Abstract

Hurricanes are devastating natural disasters which dramatically modify the physical landscape and alter the socio-physical and biochemical characteristics of the environment, thus exposing the affected communities to new environmental stressors, which persist for weeks to months after the hurricane. This paper has three aims. First, it conceptualizes potential direct and indirect health effects of hurricanes and provides an overview of factors that exacerbate the health effects of hurricanes. Second, it summarizes the literature on the health impact of hurricanes. Finally, it examines the time lag between the hurricane (landfall) and the occurrence of diseases. Two major findings emerge from this paper. Hurricanes are shown to cause and exacerbate multiple diseases, and most adverse health impacts peak within six months following hurricanes. However, chronic diseases, including cardiovascular disease and mental disorders, continue to occur for years following the hurricane impact.

## 1. Hurricanes and Health: A Conceptual Framework

Hurricanes are devastating natural disasters which take a heavy toll on lives and dismantle property and infrastructure. The economic and health impacts of a hurricane largely depend on its size and intensity and its location in terms of population density and proximity to the coastline. For example, Hurricane Katrina was one of the deadliest and costliest hurricanes on record to strike the United States, which resulted in about 1200 deaths and catastrophic damage—estimated at $170 billion USD [1].

Hurricane Maria, which impacted Puerto Rico in 2017, accounted for $94.5 billion USD in economic losses and 146 deaths [2,3]. However, these estimates can be subject to downward bias and do not fully account for indirect economic costs and disease and disability burden associated with hurricanes. For example, research suggests that the estimated death toll in Puerto Rico associated with Hurricane Maria was 2975, against 64 reported officially [3]. While the direct economic losses and loss of lives have been subject to accounting, the indirect health effects of hurricanes and their persistence over time have been challenging. Most indirect health impacts of hurricanes can be attributed to new or modified environment stressors caused by hurricanes because hurricanes change socio-physical and biochemical environmental stressors in the affected areas. To tease out the precise health impacts of hurricanes, it is important to understand direct and indirect linkages between hurricanes, environmental stressors and human health, as demonstrated in Figure 1.

Hurricanes directly modify the physical landscape and dramatically alter socio-physical environmental conditions, and these conditions deteriorate for days/weeks (e.g., flooding, water contamination, increase in air pollution) after the hurricane (landfall) before they begin to improve, thus exposing the impacted communities and displaced individuals (for example, in shelters) to new environmental stressors which can persist for weeks to months following the hurricane.

The short-term health effects of these changes in the environmental conditions are often visible within days and weeks, e.g., morbidity and mortality due to unintentional injuries, drowning, heat stress and infectious disease outbreak due to exposure to contaminated food and water. However, the long-term health impacts of hurricanes can go unnoticed. Previous literature focuses on three major themes in regard to the health impact of water related disasters, which include exposure to toxins, population susceptibility and health systems infrastructure. The cited direct health impacts include: weather-related morbidity and mortality, waterborne diseases and water-related illnesses, vector-borne diseases, and mental health effects [4]. This review attempts to explain specific rates and durations of health impacts following a hurricane, as observed in the literature. Further, the paper conceptualizes direct and indirect health impacts of hurricanes and their persistence over time. The paper also identifies research gaps in the literature and discusses methodological challenges that constrain our ability to tease out the extent of the disease and disability burden associated with hurricanes and potential areas of future research.

## 2. Methods and Materials

### Literature Review Criteria

The literature from PubMed National Library of Medicine (NLM) was reviewed using EndNote X8 (Clarivate, Philadelphia, PA, USA). The fields for search were set to “any field” and “contains.” The search criteria are listed below, starting with the most focused and moving to the broadest terms. Relevant references were not repeated if they had already been noted. Studies were considered relevant if they: (a) contained empirical data, which included the numbers of people hospitalized or affected by illness and (b) listed the name(s) of a specific storm (Table 1). The relevant studies were tabulated and the time lag between disease occurrence and the hurricane was analyzed using descriptive statistics (Table 2).

## 3. Health Impacts of Hurricane—A Review

First, we describe factors that modify adverse health effects of hurricanes. Second, we summarize disease specific effects associated with hurricanes as reported in the literature. Finally, we summarize the (duration of) persistence of the adverse health effects following hurricane landfalls.

### 3.1. Factors That Modify Health Effects of Hurricanes

#### 3.1.1. Change in Environmental Stressors

Hurricanes are shown to increase exposure to socio-physical and biochemical stressors due to physical damage to infrastructure and manufacturing facilities and flooding. New chemicals are released in the environment [5], such as releases of chemicals used in manufacturing, including petroleum, plastics, synthetics and resin [6]. Likewise, the flooding of areas with physical damage, wastewater facilities and agricultural land can result in excess run off various chemical and pathogens, which are shown to increase the risk of various diseases including infectious disease [7,8]. Hydrogen sulfide is commonly observed in the flooded areas with animal facilities. It can become airborne, which is a potential inhalation hazard [9]. Floodwaters are also found to have increased concentrations of aldrin, arsenic, lead (PB), and semi-volatile organic compounds beyond the thresholds for human health screening levels after Hurricane Katrina in New Orleans [10]. The concentration of lead found in children’s blood decreased 10 years post-Katrina due to the input of low Pb sediment residues by storm surge and the introduction of low Pb landscaping materials from outside of the city [11]. Post-flood sampling from Katrina revealed that 37% of soil samples exceeded screening guidelines for arsenic concentration, attributable to the deposition of arsenic-contaminated sediments [12].

An increase in bioaerosols, including mold and endotoxins, is most obvious in flooded areas following a hurricane landfall. The literature suggests that homes with greater flood damage after a hurricane had higher levels of mold growth [13], and elevated levels of airborne mold spores were observed in flooded homes [14]. This exposes the occupants of these homes to mold spores through inhalation. For example, following Hurricane Katrina, repair workers reported several respiratory illnesses, including sinusitis, toxic pneumonitis, and “Katrina cough”. These workers were exposed to high levels of thoracic particulate matter, endotoxins, and glucan [15]. Mold levels and respiratory symptoms associated with exposure were observed to decrease in the two months following a hurricane [16].

Exposure to airborne chemicals also increases due to the loss of electricity and post-hurricane restoration efforts including debris removal and infrastructure repair. For example, carbon monoxide poisoning is a common health risk during power outages, if generators are not operated correctly. A 283-fold increase in gasoline exposure was observed following Hurricane Sandy, which was associated with gastrointestinal and respiratory symptoms [17]. During a 9-day period after Hurricane Irma, the Department of Health (DOH) of Miami-Dade County, Florida reported 106 cases of carbon monoxide poisoning, where the most frequent symptoms included headache, dizziness and nausea [18]. Following Hurricane Ike, Texas poison control centers received 54 calls from persons exhibiting symptoms of carbon monoxide poisoning [19].

#### 3.1.2. Disruption in Healthcare Delivery

##### Interruption in Healthcare Delivery

Hurricanes dismantle infrastructure, including healthcare infrastructure, which interrupts healthcare delivery. The intensity of the damage depends on the hurricane’s intensity. Flooding and damaged transport services also limit access to healthcare facilities. Moreover, power outages and flooding force hospitals and pharmacies to close, limiting access to health services. For example, following Hurricanes Irma and Maria in the US Virgin Islands, most of the healthcare infrastructure, including hospitals and pharmacies, was damaged due to flooding, structural damage, and staff shortages [20]. In New Orleans after Hurricane Katrina, three major hospitals were surrounded by floodwater and remained closed for several months. Thus, it limits and often delays patients’ access to healthcare, especially for non-life-threatening conditions [21].

Hurricanes also disrupt other healthcare facilities, such as free-standing dialysis centers and mobile healthcare services. This occurred during Hurricane Sandy in New York City, which led to a surge of hemodialysis patients to Brooklyn hospitals. Hospital workers reported a lack of dialysis documentation, staff shortages, staff transportation, and communication with other dialysis agencies [22]. Those who evacuate may not know where to access doctors and medications in a new environment. Following Hurricane Mitch in Honduras, the availability of food, water, and healthcare dramatically decreased immediately after the storm. It took four months to restore the basic infrastructure and fully functional healthcare facilities. Lack of access to healthcare may cause additional storm-related complications. Diarrheal illness was found to be correlated with poor access to medical care in the aftermath of Hurricane Mitch [23].

##### Inadequate Access to Healthcare

There is a surge in disease burden in the aftermath of a hurricane, and timely access to healthcare is critically important. The reasons for medical care vary from storm to storm. Among patients who were surveyed during Hurricane Katrina, the main complaints included illnesses (71%), medication and refill requests (21%), and injuries (8.5%). Among those complaining of illnesses, 25% were ears, nose, and throat, 17% were dermatologic, and 11% were cardiovascular [24]. People evacuated before and after the hurricane often lack access to care. For example, in response to Hurricane Katrina, about 30,000 people were evacuated from the Gulf Coast region who resided in shelters in eight different states. However, healthcare facilities in these shelters were inadequate, especially primary and preventive care services [25]. Of those who arrived to shelters with chronic medical conditions (55.6%), 48.4% lacked medication. The most chronic medical conditions included hypertension, hypercholesterolemia, diabetes, pulmonary disease, and psychiatric illnesses [26]. The shelters for Hurricane Katrina evacuees were very crowded, which allowed infectious diseases among sheltered residents. Moreover, many patients who came to shelters presented symptoms that warranted immediate medical attention. Their symptoms included dehydration, dyspnea, injury, and chest pain [26].

##### The Lack of Access to Care Exacerbates Pre-Existing Conditions

Lack of access to care following a hurricane may have a negative impact on pre-existing medical conditions. Patients with head and neck cancer had lack of access to cancer care and treatment facilities in the aftermath of a hurricane [27]. One study found a significantly higher mean delay in chemotherapy, scheduled surgery, and radiotherapy at a city hospital for gynecologic oncology care following Hurricane Sandy [28]. Additionally, after Hurricane Maria, the gynecologic cancer service centers in Puerto Rico and the Virgin Islands were destroyed [29]. Patients were unable to access cancer treatment and care for many months after the hurricane, suggesting increase adverse health outcomes due to the lack of access to healthcare services and the delay in treatment. Additionally, many survivors of Hurricane Katrina with pre-existing chronic conditions reported that their treatment was significantly disrupted as a result of the storm [30].

##### Inequalities in Accessing Care

Existing health disparities may be exacerbated following a hurricane. Populations with impaired access to care are less likely to have access to care in the aftermath of a storm [31]. Medicare patients, as well as displaced individuals with complex medical histories, have difficulty accessing care following a natural disaster [32]. Countries with lower per capita income and resources have a relatively higher incidence of mortality following a storm [33]. Ethnic differences are reported in terms of restoring basic services, such as restoring power and healthcare facilities [34]. Likewise, socioeconomic status was found to significantly correlate with diabetes management preparedness during the hurricane [35], which exacerbates health disparities in the aftermath of a hurricane. Hurricane Katrina worsened diabetes related health disparities [36]. Similarly, disparities in Type 1 Diabetes Mellitus among children were observed in New York and New Jersey in the aftermath of Hurricane Sandy, and food disparities in African-American neighborhoods worsened in the aftermath of Hurricane Katrina [37].

##### Medical Equipment

Disruption in transport facilities also limits medical supplies and equipment. For example, shortages of sterile saline solutions were reported after Hurricane Maria in Puerto Rico [38].

#### 3.1.3. Modifying Risk Factors Exacerbate Health Risks

A host of factors can modify or exacerbate the adverse health effects of hurricanes. These factors include age, socio-economic status and pre-existing health conditions.

##### Age

The elderly population is more vulnerable to hurricanes’ direct and indirect impacts, as evident from an increase in the number of patients in the age group 40–64 following Hurricane Maria in Puerto Rico [39]. Following Hurricane Sandy, the proportion of visits for dialysis, respiratory issues, social and syncope visits was significantly higher for geriatric patients as compared to younger patients [40]. There is a disproportionately high increase in emergency department utilization by older adults, specifically those of >85 years of age. The primary diagnoses among older individuals included dialysis, electrolyte disorders, and prescription refills [41]. Moreover, older patients may already possess a health issue, which leads to their increased risk during a hurricane. For example, following Hurricane Gustav, there was a 3.9% increase in death after 90 days for residents with severe dementia who evacuated [42].

##### Pre-Existing Health Conditions

People with pre-existing health conditions, such as asthma, diabetes and cardiovascular disease, are at a greater risk of exacerbation of underlying diseases. The leading causes of excess deaths following Hurricane Maria were reported as heart disease, diabetes, Alzheimer’s disease, and septicemia [43]. Additionally, although mortality was not significantly elevated, overall morbidity increased by 12.6% following Hurricane Katrina [44]. Following Hurricane Iniki in Kauai, Hawaii, the incidence of cardiovascular disease and asthma for the 2-week period following the storm increased significantly [45]. Similarly, up to three months after Hurricanes Irma and Maria in the US Virgin Islands, a significant increase in the number of patients seen for diabetes-related and respiratory complaints, especially asthma, was reported [20]. Multiple comorbidities can further exacerbate certain health conditions. A study documented an association between disaster-related post-traumatic stress disorder (PTSD) symptoms and post-hurricane asthma attacks following Hurricane Katrina [46]. Additionally, after Hurricane Harvey, one-half of patients reported symptoms of allergies [47].

##### Socioeconomic Status (SES)

After a hurricane, an increase in emergency department visits and hospitalizations for various medical conditions is typically observed. However, there was a decrease in overall emergency department use in the most vulnerable evacuation zone immediately following Hurricane Sandy. Those who were more likely to visit emergency departments following Hurricane Sandy included: patients who were dependent on dialysis and ventilators, and homeless or elderly patients with diabetes, dementia, cardiac conditions, limited mobility, or drug dependence. These patients were more likely to develop drug-resistant infections, require isolation, and present for hypothermia or environmental exposures [48].

After Hurricane Sandy in the greater New York metropolitan area, in-patient hospitalizations for dehydration were 66% higher than in the three previous years. Hospitalizations for dehydration affected the elderly, as rates for those over 65 increased by nearly 80% following the storm [49]. Additionally, the risk of hospital visits or admissions for cardiovascular disease, respiratory disease, and injury was more than twice as high immediately, 4 months, and 23 months after the storm [50]. Additionally, emergency department treatment for respiratory diseases was observed to increase following Hurricane Sandy [51]. Medicare patients comprised a significantly higher proportion of emergency department visits for diabetes care. These visits were associated with a significantly increased frequency of comorbidities, including hypertension and chronic skin ulcers [52]. The number of emergency room visits for Type 2 Diabetes care increased by 84% during the week of Hurricane Sandy [53].

### 3.2. Disease Specific Effects of Hurricanes

#### 3.2.1. Unintentional Injuries

Direct health effects of hurricanes stem from direct wind force, damaged infrastructure and flooding. Unintentional injuries are most common among the direct health effects of hurricanes. Drowning, poisoning, electrocution and injuries from rehabilitation/restoration efforts are some of the common reasons underlying unintentional injuries. In an emergency department treating patients for injuries related to Hurricane Hugo in North Carolina, over half of the injuries were wounds and insect stings, and nearly one-third of the injuries were related to chain saw use [54]. Such an increase in unintentional injuries due to lacerations, puncture wounds, and falls was observed for over three weeks following the storm [55].

After Hurricane Sandy, tree-related injuries increased significantly [56]. Deaths after Hurricane Isabel in Virginia included drowning, traumatic head injuries from falling trees, motor vehicle crashes, and power outages. The presence of alcohol and or drugs was observed in several of the reported deaths [57]. Among patients needing orthopedic surgery, a majority of injuries were to the upper and lower extremities [58]. Among emergency department patients who were seen for hurricane-related injuries, the most common types were abrasions, lacerations, and cuts caused by falls, slips, trips, and being hit by/on an object. Most of these injury visits occurred within 3 months after the hurricane [59]. Among victims of Hurricane Katrina, the major causes of death were drowning, injury/trauma and heart conditions [60]. Electrocution was also reported as a leading cause of death [61]. Most deaths caused by drowning in Hurricane Sandy occurred within 3 days of landfall, and the majority of deaths that occurred up to one month following the storm were caused by a fall [62]. The greatest increase in unintentional injuries occurred in the third quarter following the storm, which is associated with rebuilding and recovery [63].

During the post-impact phase of a hurricane, a high rate of cuts, lacerations, and puncture wounds is observed [64]. The risk of injury/illness has been found to increase with area damage and decrease with evacuation. It is also associated with distress and disability [65]. Greater proportions of residents (in comparison to relief workers) were injured during the repopulation period following Katrina, and the leading mechanisms of injury were falling, cutting, stabbing and piercing [66]. These results show that among various hurricane events, there have been widespread injuries. Results indicate that many injuries do not occur directly after the hurricane, but instead such injuries occur during the rebuilding efforts in the post-impact phase. It is important to consider safety measures during the rebuilding phase following a hurricane to mitigate injuries and trauma.

Unintentional injuries are also observed in the first responders due to exposure to animal and insect vectors and floodwater. The literature suggests that bites from domestic animals were a top complaint of disaster rescue teams. Most bites were severe and occurred within the first 3 days following the hurricane, then gradually decreased [67].

##### Respiratory Illness

The literature suggests an increase in new respiratory-related illnesses following a hurricane. New respiratory issues were reported among the restoration workers of Hurricane Katrina in New Orleans. These symptoms included fever/cough, sinusitis symptoms, pneumonia and new-onset asthma [68]. Similar symptoms were reported among children following Hurricane Gilbert. Children reported an increase in respiratory symptoms including labored breathing, coughs, and nasal discharges in the first 2-month period following the hurricane [69]. The literature is sparse in explaining the reason underlying an increase in respiratory illnesses following a hurricane. However, a study after Hurricane Katrina suggested an association between upper respiratory symptoms reported among children four months after the hurricane. It also showed an association between roof, glass, and storm damage and home and upper respiratory symptoms [70]. This suggests that exposure to mold-spores may be directly responsible for increased self-reports of upper and lower respiratory symptoms. The duration of respiratory illness is not well-cited in the literature. However, one study suggests that over the course of a 12-month assessment, asthma symptoms in children decreased [71].

##### Chronic Disease

The lasting health impacts of a hurricane have been observed to exacerbate pre-existing chronic conditions. For example, among diabetic patients in New Orleans, a significant difference between pre- and post-Katrina conditions was indicated by hemoblogin A1C (HbA1c,) high-density lipoprotein (HDL), and systolic and diastolic blood pressure values [72]. Chronic renal failure was also exacerbated after Hurricane Katrina. For example, those on a transplant list had to wait longer for an organ donor, as mass evacuation overwhelmed dialysis centers, there was a lack of availability of medical records, and the availability of organs for transplantation was limited since communication and transportation lines became severed [73].

##### Reproductive Health

Several studies have examined the effects of hurricanes on pregnancy outcomes. In New York during the month of Hurricane Sandy, there was an increase in the number of pregnancy complications leading to emergency department visits. This suggests that pregnancy complications are more likely to occur following a hurricane. Additionally, seven to eight months after the storm, gestational hypertension and renal disease occurred at higher levels [74], suggesting stress from the storm as one of the reasons underlying adverse health outcomes among pregnant and postpartum mothers. Other pregnancy complications cited in the literature include preterm deliveries and fetal mortality. Women in areas affected by hurricanes are more likely to experience preterm deliveries [75]. One study found that every 1% increase in the destruction of housing stock from Hurricane Katrina was associated with a 1.7% increase in fetal death [76].

Postpartum mental health issues have been commonly reported among women who were pregnant during or shortly after a hurricane. Among Hurricane Katrina surveys, at two months postpartum, 18% of women met the criteria for depression and 13% for PTSD. Increased risk for both depression and PTSD resulted from severe storm experiences [77].

Women require gender-specific medical care. Difficulties accessing gynecological care after a hurricane may lead to unintended health consequences as well. For example, after Hurricane Ike in Texas, women reported difficulty accessing contraception. Lack of access to birth control was associated with a higher frequency of unprotected sex among women [78]. This suggests that the lack of access to contraception, caused by lessened availability of medical care during a hurricane, may lead to an increased frequency of unintentional pregnancy or sexually transmitted infections.

Rates of sexually transmitted infections may also increase following a hurricane as testing centers close. HIV testing is one of the primary methods of disease prevention. However, following Hurricane Sandy, it took nearly 17 weeks for testing levels to return to normal in high-impact areas [79]. This could have caused increased rates of HIV infection. One study, however, has found no relationship between exposure to hurricane weather and reproductive health among women [80].

##### Mental Health

There is an abundance of literature regarding the effects of hurricane exposure on mental health. Increased visitation for mental illness was observed following Hurricane Sandy, with a peak in visits occurring eight months after the storm [74]. A surge in psychiatric emergency department visits, which persisted for 4–6 months, was observed following Hurricane Sandy. However, the surge in psychiatric hospitalizations was only observed for 1–3 months after the storm made landfall [81]. The number of psychiatric visits increased, most notably in the first month after Hurricane Sandy. The percentage of admissions declined, while the average length of stay declined [82]. However, these studies do not quantify the duration of these health effects.

Elevated symptoms of PTSD, depression, and anxiety are cited as the most common mental health impacts of hurricanes [83]. Higher than average levels of PTSD and depression symptoms were reported among Hurricane Katrina survivors [84]. Risk factors for having a PTSD diagnosis include expressing a fear of death, witnessing the injury or death of a loved one, seeing violence, and directly experiencing violence themselves [85]. PTSD and depression symptoms were also reported following Hurricane Sandy [86] and Hurricane Mitch [87]. Other studies have identified risk factors for PTSD, anxiety and depression. Following Hurricane Katrina, self-reported flooding intensity was positively associated with anxiety, depression and PTSD [88]. Additionally, another study found that experiencing one or more natural disasters (fire, tornado, flood, earthquake, or hurricane) by age five is associated with an increased risk of mental health disorders, specifically anxiety disorders, in adulthood [89]. These results suggest that increased exposure to natural disasters and environmental hazards from hurricanes may be associated with anxiety, depression and PTSD. These results are further confirmed by a study that found that one year after Hurricane Katrina, there was a significant association with having PTSD and staying in New Orleans instead of evacuating [90]. Other risk factors for elevated levels of PTSD one year post-Katrina included having material losses, experiencing the death of a friend or family member, needing healthcare during the storm, and not having access to healthcare needs during the storm.

Several studies quantify the lasting impacts of hurricanes on mental health. One study concluded that fifteen months after Hurricane Katrina, almost half of New Orleans residents continued to experience poor mental and physical health [91]. Among children who survived Hurricane Katrina, mental health issues persisted up to four years after the disaster [92]. This result suggests that mental health, in relation to hurricane impacts, is long-lasting and widespread. Additionally, it suggests that post-disaster mental health services and case management should remain available for years after the event. However, following Hurricane Sandy, there were statistically significant decreases in anxiety scores and PTSD scores between 1-year of follow-up and baseline [93], suggesting that mental health issues related to the storm did not persist for greater than one year. Some predicted that suicidal ideation would increase following a hurricane. However, in a survey employed post-Katrina, suicidal ideation was lower after Katrina due to a strong relationship between dimensions of personal growth after trauma [94].

The relationship between mental health impacts and substance use following a hurricane has been investigated. Following Hurricane Rita in Louisiana, there was a significant positive relationship between PTSD symptoms and increases in alcohol and marijuana use at 7-months follow-up [95]. Additionally, exposure to hurricane-related traumatic events following Hurricanes Katrina and Rita was associated with higher odds of binge drinking [96]. This suggests that the mental health impacts of a hurricane may lead to other health risks, such as increased drug and alcohol use.

##### Violence

People experience stress during and after hurricanes. In some cases, this can manifest as intimate partner violence [97,98]. An association of increased interpersonal violence following a hurricane with daily stressors and loss of control is reported in the literature [99,100]. Reports of intimate partner violence increased significantly following Hurricane Katrina. A greater percentage of women reported both psychological abuse and physical victimization [101].

Violence against children has also been reported. One study suggests that the incidence of inflicted traumatic brain injury in children increases following a hurricane [102]. After Hurricane Ike in Texas, an increase in physical and sexual child abuse was associated with poor sleep quality [103].

##### Infectious Disease

Outbreaks of infectious disease in the aftermath of a hurricane have been commonly reported. The etiology of these diseases may be bacterial or viral. According to the literature, infections including skin and soft tissue infections, gastrointestinal infections, respiratory infections, zoonotic infections, and vector-borne diseases pose a threat after hurricanes, specifically to vulnerable and displaced populations [104]. Among those displaced during Hurricane Katrina, many were forced into over-crowded shelters, where disease outbreaks could easily spread. Following Hurricane Sandy in New York, several outbreaks of infectious viral gastroenteritis were reported in shelters [105].

Floodwater has been associated with bacterial outbreaks. For example, after Hurricane Sandy, a case of legionellosis was reported in association with water-impacted areas. However, overall, Hurricane Sandy did not increase the disease prevalence in New York City [106]. While this may have been true for this particular storm, other storms have seen a significant increase in infectious diseases. After three recent hurricanes in Louisiana, an increased number of residents tested positive for nontuberculous mycobacteria (NTM) [107], suggesting an association between the increased NTM infection and hurricanes.

After Hurricane Matthew in Haiti, a significant increase in cholera cases was reported. Cases began to decline 46 days after the storm made landfall [108].

Floodwaters and still-water provide habitats for mosquito breeding. Following one hurricane, this caused mosquito abundance to increase. An increase in mosquito abundance is associated with an increased probability of the spread of vector borne diseases [109]. For example, the risk of Zika cases also increased in Haiti after Hurricane Matthew [110]. However, mosquito concentrations may also decrease following a storm. After a hurricane in Belize, the mosquito abundance dropped, which led to a declined malaria risk [111].

Floodwaters may also exhibit increased bacterial concentrations. Following Hurricane Katrina, restoration workers reported skin rashes. The most common cause of rash was popular urticarial, which was associated with sleeping in previously flooded huts [112]. This suggests that floodwaters may lead to bacterial infection. In addition, fecal coliform bacteria may occur in elevated concentrations in surface floodwater following a hurricane. This was observed after Hurricane Katrina [113], as well as in the Florida Keys following Hurricane Irma [114].

### 3.3. Health Impact Persistence

The persistence of the adverse health impacts of hurricanes varies by disease types and storm intensity. Unintentional injuries due to hurricane wind force and flooding occur within days and weeks. For example, emergency room visits returned to baseline levels within two weeks after Hurricane Harvey, and the highest frequency of emergency department visits was clustered within five counties with disaster declarations during an 11-day window [115]. Chronic diseases associated with changes in socio-physical and biochemical stressors and an interruption in healthcare delivery also occur immediately after the hurricane and can persist for months. For example, chronic disease treatment accounted for 33% of hospital visits, and the number of chronic-disease related visits peaked ten days after Katrina made landfall [116]. Hospitalization rates of dialysis patients also significantly increased in the month after Hurricane Katrina made landfall [117]. Likewise, following the 2004 hurricane season in Florida, most storm-related deaths resulted from indirect impacts of the storm. Specifically, cardiovascular diseases, cancer, diabetes, and accident-related deaths were the leading causes of mortality. Elevated mortality was observed for up to two months after each storm [118].

After Hurricane Katrina, there was a more than 3-fold increase in the percentage of admissions for acute myocardial infarction (AMI) observed for up to six years following the storm. Patients admitted for AMI were observed to have significantly higher rates of psychiatric comorbidities, smoking, lack of health insurance, and unemployment [119]. Hospitalizations for cardiovascular disease also increased directly after Katrina. Hospitalization rates peaked on the sixth day following the landfall, and returned to pre-hurricane levels after two months [120]. Chronic health impacts of hurricanes can persist for a longer duration. A study shows that 25–40% of those who lived within the Gulf Coast region affected by Hurricane Katrina and Hurricane Rita are living with a chronic illness. Other studies also report that the prevalence of emergency department visits and hospitalizations remained significantly elevated during the entire year after the storm [44].

To assess the persistence of the adverse health impact of hurricanes, we reviewed 50 studies that included empirical data on the disease-specific effects associated with hurricanes. Of these 50, 19 reported two or more diseases, but we tabulated these data by disease type. Thus, the final analysis included a total of 93 disease records. However, only 29 of these 93 records reported time lag between disease reporting and hurricane/storm landfall time. For the rest, we used the difference between study data collection time and the date of hurricane/storm landfall, which resulted in 67 disease-specific records with the quantifiable time-lag. These records were classified under 11 disease categories (Table 2). More than one third (37 of 93) of reported diseases concerned Hurricane Katrina (landfall in 2005), followed by Hurricane Sandy (19 of 93) (landfall in 2011).

Unintentional injuries and cardiovascular diseases were most frequently (16.4% each) reported, followed by infectious diseases, respiratory diseases and mental-health disorders (10.4% each). About one quarter (25.4%) of the diseases occurred within three months following the hurricane. Disease occurred most frequently within 3 and 6 months following the hurricane. About 44% of diseases occurred after a year. As evident from Table 1, the persistence of adverse health impacts also varies by disease type (χ^2^ ~ 83.7; *p* ~ 0.002). For example, unintentional injuries occurred within six months after the hurricane, and chronic diseases, including respiratory and cardiovascular diseases, occurred between 3 and 6 months following the hurricane.

## 4. Discussion

Adverse health effects associated with hurricanes have been subject to research scrutiny. A majority of studies (60%) examined different health effects associated with two hurricanes: Katrina and Sandy, which struck in 2005 and 2011, respectively. The adverse health effects of hurricanes last for months and years. The literature may inadequately represent the extent of disease burden associated with all hurricanes which have struck since 2000, including a series of hurricanes in the US since 2017. First, most studies focused on more severe storms and limited research was available on category 2 or 3 hurricanes. Second, most studies examined the health impacts of hurricanes at one point in time and very limited studies conducted long-term follow up. Third, the most obvious life-threatening diseases, such as unintentional injuries and cardiovascular diseases, were frequently studied. For example, only one study examined cancer related deaths due to hurricanes [118]. However, the severity of cancer can increase due to the lack of access to chemotherapy [28]. Fourth, the literature suggests that pre-existing health conditions worsen the severity of different diseases [43,44,45], but it sheds little light on the onset of new chronic diseases, such as asthma, allergies, diabetes and mental disorders. Finally, only a few studies examined the long-term health impacts of hurricanes [121].

As conceptualized in Figure 1, there is an intricate direct and indirect relationship between hurricanes and health. This makes it challenging to tease out the extent of disease and disability burden associated with hurricanes, especially the indirect and long-lasting adverse disease burden, because environmental conditions deteriorate with the landfall of a hurricane and new environmental socio-physical and biochemical stressors persist for weeks to months following a hurricane. For example, there is an increase in allergens, especially mold in water damaged buildings, and air pollutants, including airborne particulate matters due to restoration efforts, cleaning debris, the use of generators in the absence of electricity, etc., and an increase in the severity of asthma, chronic obstructive pulmonary disease (COPD) and other respiratory diseases. The literature sheds little light on environmental mediated health impacts of hurricanes. Many studies documented an increase in mold due to flooding but failed to show a direct link between mold and associated morbidity, such as respiratory diseases and allergies [13,14,15]. Likewise, studies document food desert, but fall short on establishing the link between nutrition and associated disorders, including disease of metabolic syndrome. Health impacts are amplified because of an abrupt and simultaneous increase in exposure to multiple stressors, e.g., heat stress due to loss of electricity, lack of access to healthcare and clean water, poor or under-nutrition, loss of property and resources, and the exacerbation of existing health conditions [122].

Uncertainty and underreporting of the diseases and disability attributable to hurricanes warrant a critical assessment of the data and methodological gaps needed to tease out the precise disease burden of hurricanes. Location-specific longitudinal data on the intensity of the hurricane impact, health outcomes and changes in socio-physical and biochemical stressors are needed to quantify the complete disease burden of hurricanes. There are three potential data sources that can be used to assess the health effects of hurricanes: (a) vital records—births and deaths, (b) hospital visit records and (c) survey-based screening of the affected communities. Each of them has its inherent advantages and disadvantages, but they complement each other. Vital records are generally used to examine adverse birth outcomes and mortality associated with hurricanes. However, these data may not include underlying causes for all deaths (e.g., the classification of diseases may be missing for many cases), making it challenging to tie them to hurricanes. Hospital records are used extensively to determine hurricane related morbidity. These data are time-stamped and have precise disease classification. However, these data can be subject to bias, especially for non-life-threatening conditions, because non-emergency services are interrupted for weeks to months following a hurricane and people do not seek care for non-life-threatening conditions. Studies based on hospitalization data can also be subject to bias. For example, a patient with cardiovascular disease who was hospitalized after the hurricane due to the interruption in healthcare services may not need to be hospitalized once the regular healthcare services are restored. Screening the affected communities for different diseases (using structured survey instrument and biospecimen collection) can be used to assess underreported diseases. However, such screening is required at incremental time intervals to gauge the time lag in the occurrence of diseases and a hurricane landfall. In addition, questions concerning health and disease status in pre- and post-hurricane periods can be used to determine the onset of new diseases.

A standard methodology used to quantify the health effects of hurricanes is to determine excess mortality and morbidity from baseline, i.e., the prevalence of disease under normal conditions. However, such a comparison does not factor in interruption in healthcare services due to closure and/or physical damage to facilities, mediational and moderating impacts of environmental stressors magnified by hurricanes and those displaced or migrated (permanently) from the impacted areas. Moreover, some of the adverse health effects, such as the onset of allergies or mental disorders, may not surface until many months after the hurricanes. Practitioners, public health officials, and emergency management should be aware of the potential of such adverse health outcomes following hurricanes.

## 5. Conclusions

The increasing frequency and intensity of hurricanes in recent years pose an unprecedented threat to the public health of communities living in coastal areas. As the literature suggests, a myriad of diseases are tied to hurricanes. The precise quantification of disease burden and its healthcare cost associated with high- and low-impact hurricanes is warranted to guide hurricane preparation, mitigation and adaptation strategies. This paper suggests the use of longitudinal data sets from complementary sources (vital records, hospital records and community screening) to gauge the precise burden of diseases and identify new diseases and indirect and long-lasting health impacts associated with hurricanes. Although this paper highlights the health impacts of hurricanes and their persistence over time, this paper has implications for preparing, mitigating and managing the overall impacts of natural disasters.

## Figures and Tables

**Figure 1 ijerph-18-02756-f001:**
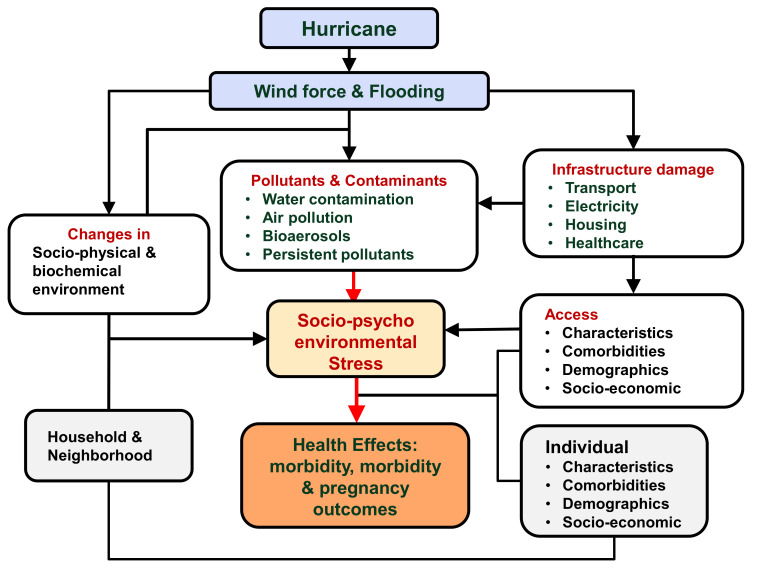
A conceptual model of the direct and indirect health effects of hurricanes.

**Table 1 ijerph-18-02756-t001:** The literature search key terms, results and relevant references.

1st Search Word	2nd Search Word	3rd Search Word	Number of References	Number of Relevant References
Hurricane	Health	Impact	386	24
Hurricane	Health	Impacts	103	2
Hurricane	Health		2063	21
Hurricane			4126	3

**Table 2 ijerph-18-02756-t002:** Disease frequency by time-lag between disease occurrence (or reporting) and date of hurricane impact (or landfall) (percentages in parenthesis).

Disease	Time Lag (Month)						Total *n* (%)
	1	1–3	3–6	6–12	12–18	18–36	
Cancer	0	0	1	0	0	0	1 (1.5)
Cardiovascular	0	2	4	0	0	5	11 (16.4)
Diabetes	0	0	3	2	0	1	6 (9.0)
Environmental disease	1	1	3	0	0	0	5 (7.5)
Infectious	1	4	2	0	0	0	7 (10.4)
Unintentional injury	2	4	4	0	1	0	11 (16.4)
Mental Health	1	0	0	0	2	4	7 (10.4)
Reproductive Health	0	0	0	0	1	3	4 (6.0)
Respiratory Health	0	1	4	1	1	0	7 (10.4)
Substance abuse	0	0	0	0	1	2	3 (4.5)
Violence	0	0	0	2	3	0	5 (7.5)
Total (%)	5 (7.5)	12 (17.9)	21 (31.3)	5 (7.5)	9 (13.4)	15 (22.4)	67 (100.0)

values in parenthesis are percentages.

## Data Availability

The data used in this paper were compiled from publically available data sets. However, if any one needs the compiled data, these data can supplied upon a reasonable request to the corresponding author.

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
