# Peer review of "Perspectives on the Health Effects of Hurricanes: A Review and Challenges"

_ijerph, 2021, doi:10.3390/ijerph18052756_

Round 1

Reviewer 1 Report

I believe that "Perspectives on the Health Effects of Hurricanes" is a very important article and it merits publication in your important journal.

The topic is important for several reasons.  First, hurricanes are fairly frequent (each year) in coastal areas.  Second, they do generate significant impacts and these are described in detail by the authors.  I commend them for a very clear discussion.  Third, to my knowledge the health consequences have not been sufficiently studied.  Fourth, this study has several implications for vulnerable populations and the medical professionals and institutions that serve them.  And, finally, this piece admonishes scholars to be very cautious about research methods and findings (since studies often rely on just a few major hurricanes).  Therefore, I believe this study deserves to be published.  In fact, as someone who studies disasters and emergency management, I wish I would have seen this type of research long ago.

There are a few recommendations to improve the paper:

  1. Should there be a subtitle that discusses research methods?  This would be logical since the discussion really highlights the methodological issues surrounding prior studies.
  2. The authors are generally correct that hurricanes are devastating.  However, some scholars may state that earthquakes are more deadly and destructive.
  3. Perhaps add "the" before the final paragraph on page 6.
  4. If desired, the authors could note somewhere that this research is relevant to not only those interested in public health and medicine, but disaster scholars and emergency managers as well.
  5. The authors could also mention that practitioners should be increasingly aware of these health consequences after a disaster.  For instance, more planning should take place before Hurricanes to address their unique consequences.  So, this study will benefit not only scholars interested in research methods, but also emergency management and public health organizations.
  6. The conclusion could be strengthened to end on a very solid note.

Overall, a great contribution.  I believe it should be published.

Author Response

We are very thankful for your constructive feedback and appreciate the time you spent reviewing this paper.

We have added a subtitle titled “A review and challenges” to indicate this study was conducted via literature review, as you suggested. You are correct that some may state that other forms of natural disasters are more devastating, therefore we have corrected the introduction to simply state that hurricanes are devastating natural disasters. We have accepted your suggestion regarding the audience of the article, and added a section to address this in the discussion on page 16. We have also restated this information in the conclusion, see page 16. 

Reviewer 2 Report

It appears that this article is a literature  review mainly on two hurricanes (60  %) and others in the same region.This focus should be said in the title, the summary, the analysis and the conclusions.

The description of health impact and the epidemiological factors is very well done and quite exhaustive , with the time - lagged effect.

The discussion reveals well the limiting methodological factors of the publications. It would have been useful to point out the institutional context of data collection and the links with the decision making process related to hurricane preparation, mitigation and adaptation. Even it is not the purpose of this review , any more action oriented analytical assessment would be welcome

I suggest to look at WHO and AMRO guidelines on standardized information systems for hurricanes and natural disaster management

Author Response

This article is a literature review regarding any hurricane health impacts reported in Pubmed, however the majority of the literature focuses on two hurricanes (Hurricane and Katrina). We aim to demonstrate that more research regarding adverse health outcomes of hurricanes in other regions is necessary, and therefore do not specify these two hurricanes in the title. We thank you for your contributions and constructive feedback.